# Seroprevalence of SARS-CoV-2 infection in Cincinnati Ohio USA from August to December 2020

**Greg Davis**[1☉], **Allen J. York**[1☉], **Willis Clark Bacon**[1☉], **Suh-Chin Lin**[1], **Monica Malone McNeal**[1], **Alexander E. Yarawsky**[1], **Joseph J. Maciag**[1], **Jeanette L. C. Miller**[1], **Kathryn C. S. Locker**[1], **Michelle Bailey**[2], **Rebecca Stone**[2], **Michael Hall**[2], **Judith Gonzalez**[2], **Alyssa Sproles**[1], **E. Steve Woodle**[3], **Kristen Safier**[1], **Kristine A. Justus**[1], **Paul Spearman**[1], **Russell E. Ware**[1], **Jose A. Cancelas**[2], **Michael B. Jordan**[1], **Andrew B. Herr**[1], **David A. Hildeman**[1], **Jeffery D. Molkentin**[1]*

1 Department of Pediatrics, Cincinnati Children's Hospital Medical Center, University of Cincinnati, Cincinnati, Ohio, United State of America, 2 Hoxworth Blood Center, University of Cincinnati College of Medicine, Cincinnati, Ohio, United States of America, 3 Department of Surgery, University of Cincinnati College of Medicine, Cincinnati, Ohio, United States of America

☉ These authors contributed equally to this work.
* jeff.molkentin@cchmc.org

**Data Availability Statement:** All relevant data are within the paper and its Supporting information files.

## Abstract

The world is currently in a pandemic of COVID-19 (Coronavirus disease-2019) caused by a novel positive-sense, single-stranded RNA β-coronavirus referred to as SARS-CoV-2. Here we investigated rates of SARS-CoV-2 infection in the greater Cincinnati, Ohio, USA metropolitan area from August 13 to December 8, 2020, just prior to initiation of the national vaccination program. Examination of 9,550 adult blood donor volunteers for serum IgG antibody positivity against the SARS-CoV-2 Spike protein showed an overall prevalence of 8.40%, measured as 7.56% in the first 58 days and 9.24% in the last 58 days, and 12.86% in December 2020, which we extrapolated to ~20% as of March, 2021. Males and females showed similar rates of past infection, and rates among Hispanic or Latinos, African Americans and Whites were also investigated. Donors under 30 years of age had the highest rates of past infection, while those over 60 had the lowest. Geographic analysis showed higher rates of infectivity on the West side of Cincinnati compared with the East side (split by I-75) and the lowest rates in the adjoining region of Kentucky (across the Ohio river). These results in regional seroprevalence will help inform efforts to best achieve herd immunity in conjunction with the national vaccination campaign.

## Introduction

Severe acute respiratory syndrome coronavirus 2 (SARS-CoV-2), the agent responsible for Coronavirus disease 2019 (COVID19) was first identified in Wuhan, China in late 2019 [1,2], which thereafter spread across the globe resulting in the current pandemic and more than 29 million documented cases of infection in the Unites States of America (USA), of which over

**Funding:** This study was supported the Howard Hughes Medical Institute (to J.D.M. there is no grant number for investigator awards by the HHMI). J.D.M was also supported by an internal grant from Cincinnati Children Research Foundation to conduct these studies on COVID-19 (Grant #1). Contributions to this work by P.S. were financially assisted by The John Hauck Foundation. The funders had no role in study design, data collection and analysis, decision to publish, or preparation of the manuscript. https://www.cincinnatichildrens.org/ https://fconline.foundationcenter.org/fdo-grantmaker-profile?key=HAUC002 https://www.hhmi.org/.

**Competing interests:** he authors have declared that no competing interests exist.

985,000 from Ohio as of March of 2021 [3]. Current efforts to curtail the pandemic are largely dependent the national vaccine program, with the goal of achieving herd immunity [4,5]. Another consideration in achieving herd immunity is quantifying background rates of previous infection within the population [4]. In general, previously infected individuals are resistant to new infection for a period, and/or they appear to have a reduced severity of disease if re-infected [6].

SARS-CoV-2 contains a large surface facing glycoprotein called Spike (S, ~190 kDa) that facilitates binding of the virus to the receptor angiotensin converting enzyme 2 (ACE2) on host cells, which after proteolytic cleavage of S facilitates viral cellular involution and infection as previously described [7,8]. The receptor-binding domain (RBD, 30 kDa) is part of the S protein that directly interacts with ACE2, and antibodies against the RBD region can mediate neutralization and protection from viral infection [7,8]. The S protein is also a primary component of immunogenicity for the host response against the virus that produces immunity [7]. Thus, it is not surprising that a primary strategy for the vaccine involves using the SARS-CoV-2 S protein to generate an immune response [7,8].

Rates of seropositivity for SARS-CoV-2 in voluntary blood donors in the Greater Cincinnati Metropolitan Area (GCMA) of Ohio USA were examined from August 13th—December 8th of 2020, just prior to the beginning of the nationwide vaccination program. A modified serological enzyme-linked immunosorbent assays (ELISA) assay developed by Krammer and colleagues [9,10] was implemented to examine 9,550 individuals of age rage 16–91 years old for SARS-CoV-2 S protein IgG antibodies to quantify rates of prior infection in the GCMA. Rates of S protein seropositivity in the GCMA were examined based on race, gender, geographic subregions and time, which we also compared against rates of past infection with the 4 endemic human cold-causing coronaviruses (hCoV-229E, -NL63, -OC43, and -HKU1) [11,12]. The results suggest a rate of past SARS-CoV-2 infection in the GCMA that is more than 2X the rate of verified infection by PCR molecular detection.

## Materials and methods

### Human samples

10514 samples were collected from donors from August 13th through December 8th of 2020, which were remnant materials available after clinical work was completed. Samples were processed within 7 days of collection. As part of the clinical protocol, blood samples in EDTA tubes, were collected according to U.S. Food and Drug Administration (FDA) regulations and American Association of Blood Banks (AABB) guidelines. Also, as part of the clinical protocol, information on the medical, social, behavioral, and travel history of the donor was obtained. Prior to use in research, all identifying information was removed from the samples and questionnaires. Because of this de-identification, the University of Cincinnati Institution Review Board (IRB) reviewed the proposed SARS-CoV-2 serology initiative and classified it as non-human research. Donors are subjected to medical, social, behavioral, and travel history questionnaire to reduce risk of communicable diseases. Donors who felt unhealthy including, but not limited to, elevated temperature, low blood hematocrit or signs of respiratory infection were excluded. Samples without complete biogeographical information were subsequently excluded, resulting in the 9550 total samples that are reported here when also accounting for donors that gave blood more than once.

### Ethics and reporting

Blood samples were collected from volunteer donors presenting to the Hoxworth Blood Center following U.S. FDA regulations and American Association of Blood Banks (AABB) guidelines

with a signed standard donor consent form. Specimens were de-identified and as such, the University of Cincinnati Institutional Review Board (FWA #: 000003152) ruled that these blood donor samples and their analysis as constituting non-human research for the proposed study of SARS-CoV-2 serological responsiveness. There was no animal research in this report.

## ELISA

The ELISA protocol was adopted from 2 reports in the literature [9,10]. Briefly, SARS-CoV-2 antigens for S protein and RBD were coated on 96 well plates (Corning 9018) in 1X PBS at 1.0 μg/ml in 50 μl per well for S protein and 2 μg/ml for RBD protein. The antigen plates were washed (5 times with 1X PBS + 0.1% Tween-20 (PBST)) and then blocked with 3% non-fat dry milk in PBST for 1 hour at room temperature. Plasma samples at 1:100 final dilution were added to a final volume of 50 μl per well in 96-well plates [9,10]. Samples received from Hoxworth Blood Center in EDTA anticoagulated tubes were heat inactivated at 56°C for 20 minutes. Controls on each plate consisted of a plasma sample with known high S protein antibody levels. After washing 5 times using a BioTek plate washer ELx405, plates were blotted to remove all liquid and then 50 μl of goat anti-human IgG conjugated to horseradish peroxidase (HRP) (Jackson ImmunoResearch 109-035-008) in PBST was added at a dilution of 1:10,000 for 1 hour at room temperature. Plates were washed 5 times with PBST and once with 50 mM citric acid phosphate buffer, pH 5.0. The colorimetric reagent specific for HRP activity assessment, o-phenylenediamine (OPD, Sigma P4664), was added in water to the plates for 15 minutes at room temperature and the reaction was stopped with the addition of 1 M $H_2SO_4$. Spectrophotometric based absorbance at 492 nm was assayed in a BioTek Synergy 2 plate reader. Negative control serum samples from 60 individuals were used to establish the absolute baseline value for the S protein ELISA and 53 individuals for the RBD protein ELISA, and 3 times the standard deviation was summated to this average negative value in assigning a positive value threshold.

## Protein production and purification

RBD and S proteins were produced by using mammalian expression plasmids [9,10] that were transiently transfected into expiCHO™ cells (ThermoFisher, A29133) via manufacturer's instructions. Briefly, expiCHO cells were transfected with plasmid DNA (1 μg/ml of cell volume) at $6x10^6$ cells/ml in suspension culture using the Expifectamine reagent (ThermoFisher, A14525). Transfected expiCHO cells are then cultured per the manufacturers 'max titer' protocol at 32 degrees shaking at 125 rpm for 12 days. Cell culture supernatants were harvested and filtered through a 0.2 μM membrane and both S protein and RBD were purified using a 20 mL $Ni^{2+}$-charged HiPrep IMAC FF 16/10 column (Cytiva) to bind the His-tagged region engineered into each protein [9,10]. A 10 kDa MWCO centrifugal filter unit (Amicon, ACS501024) was used to concentrate fractions containing RBD. Protein purity was validated by SDS-PAGE and western blotting using a PENTA-His antibody (Qiagen, ID:34660). Purified RBD and S proteins were characterized by sedimentation velocity analytical ultracentrifugation using a Beckman Coulter XL-I. Data were analyzed using SEDFIT's continuous c(s) distribution model [13], SEDANAL version 7.45 [14], or DCDT+ version 2.4.3 [15]. Purified protein was stored at -20°C in 50% glycerol with 5 mM sodium azide.

## Luminex

Luminex assays were performed with the One Lambda COVID Plus kit according to manufacturer's instructions (ThemoFisher, LSCOV01). Briefly, the diluted plasma/serum samples and controls from the ELISA screen were combined in a 96-well MultiScreen filter plate (EDM

Millipore, MSVN1B50), 2 μl of serum/plasma was added to 17 μl of 1X PBS and then 1 μl of 0.02 M EDTA was added for a total volume of 20 μl (final serum dilution 1:10, but effectively 1:100 final dilution for reading). The plates were washed with 150 μl of wash buffer, then 100 μl of PE-conjugated anti-human IgG (One Lambda, LS-AB2) was added at 1:100 and incubated and washed again. The plates were loaded into the Milliplex 200 with reporter laser 532 nm/classification laser 635 nm for analysis (Luminex). Data from the instrument were prepared and analyzed using Bio-plex manager 6.1 software.

## Statistics

Means and Standard Deviations were determined for all data sets using Microsoft Excel Data Analysis Descriptive Statistics tool. Statistics between groups were calculated using the Microsoft Excel Data Analysis t-Test: Two-Sample Assuming Unequal Variances. All z scores and the number of standard deviations from the mean of the reference population were calculated for the difference between rates of two data sets and subsequent p-value calculated using Microsoft Excel.

## Results

Blood samples from volunteer donors through the Hoxworth Blood Center in the GCMA were collected and analyzed for antibodies against the S protein by ELISA, and positives were also analyzed for RBD antibodies in a separate ELISA. Exactly 10514 samples were collected and processed from August 13th through December 8th of 2020, which represented 9550 unique donors geographically located within the GCMA, and we determined that 802 were positive for S protein antibodies, for an overall prevalence of 8.40%. The accuracy of this reported rate of past infection is dependent on how the laboratory ELISA was implemented and verified (see Methods). The ELISA was based on a protocol described by Krammer and colleagues [9,10], which was granted Emergency Use Authorization by the United States Food and Drug Administration. However, an alternate protein production and quality control system based in expiCHO cells was used to generate S and RBD proteins (S1 Fig). Importantly, the ELISA-based reactivity of S protein antigen generated in expiCHO cells was very similar to S protein generated in expi293 cells as used by Krammer and colleagues [9,10]. The ELISA positivity cut-off was set as 3-standard deviations above background, calculated with negative serum samples from 2019 before the onset of the pandemic. The actual experimental value from the ELISA was 0.4039 for S protein IgG antibody and 0.4826 for RBD as optical density (OD) units (Table 1 and Fig 1).

All 802 donors with antibodies against the S protein were also evaluated for antibodies against the RBD in separate ELISAs, which showed that 446 were positive for both, a rate of 55.61% (Table 1). However, because the RBD region is ~85% smaller than the entire S protein, there is reduced sensitivity of the RBD ELISA due to fewer immunogenic epitopes and likely why only ~56% concordance was observed (Table 1). Indeed, a sub-analysis of the 802 positive donors showed that the highest 1/3rd OD values (2.0–3.0) had 97.3% positivity for RBD, while the lowest 1/3rd of OD values (0.4039–1.0) had only 21.3% positivity for RBD (Table 1). Thus, those donors with the lowest levels of total S protein antibodies were less likely to have detectable RBD antibodies simply based on sensitivity associated with total antigenicity between S and RBD proteins.

A Luminex immunodetection platform was also employed to examine rates of positivity against the endemic 4 human cold-causing coronaviruses (hCoV), as well as to confirm the validity and sensitivity of the S protein ELISA (Table 2). We selected a group of 11 highly positive donors for S and RBD proteins by ELISA for comparative analysis in the Luminex

**Table 1. Mean S and RBD raw ELISA OD values for setting the assay baseline and the resulting 802 positive samples separated by assay reactivity.**

|  | Mean of negative controls | Standard deviation | Mean + 3 standard deviations |
|---|---|---|---|
| Spike OD | 0.1284 | 0.09185 | 0.4039 |
| RBD OD | 0.1482 | 0.1115 | 0.4826 |
|  | # of Spike Pos | # of RBD Pos | % RBD and Spike positive |
| All positives: | 802 | 446 | 55.61 |
| Spike OD range |  |  |  |
| 0.4039 to 1.000 | 409 | 87 | 21.27 |
| 1.001 to 2.000 | 205 | 176 | 85.85 |
| 2.000 to 3.000 | 188 | 183 | 97.34 |

The final ELISA raw data for antibody reactivity against Spike (S) or RBD proteins, as measured in OD values at 492 nm in an automated spectrophotometric plate reader. The top part of the table shows how negative control serum samples were used to set the assay detection limit. The bottom part of the table shows 802 donors as positive for S protein antibodies that were also assessed for RBD ELISA antibody positivity, and while only 55.61% were positive for both S and RBD, this rate depended on the strength of the S protein antibody levels over the 3 different S protein reactivity OD ranges shown in the table.

platform, which showed 100% correlation for S and RBD from SARS-CoV-2, but not S protein from MERS-CoV or SARS-CoV-1 (Table 2). We also examined Nucleocapsid (N) protein reactivity in these same S and RBD positive samples and all were positive (Table 2). However, there was no correlation between S and RBD positivity and S1 protein positivity specific to the four cold-causing coronaviruses (hCoV-229E, -NL63, -OC43, and -HKU1) (Table 2). Eleven highly characterized negative controls from the S protein ELISA were also examined and these same samples were 100% negative for S protein, RBD and N protein in the Luminex assay (Table 2). We also analyzed 57 donor samples that were RBD negative and relatively low in overall S protein reactivity, and in this group 67% confirmed in the Luminex assay for S protein, and 23% were positive for RBD, but only one sample was positive for N protein (Table 2). Finally, while many donors showed antibodies against the 4 hCoVs in all three of our S protein ELISA groups (11 negatives, 11 high positives, and the 57 low positives), there was no

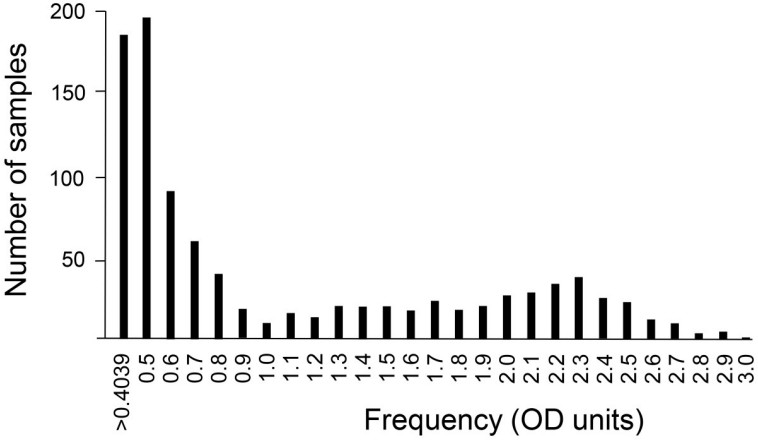

**Fig 1. Histogram frequency of positive S protein ELISA OD values from 0.4039 to 3.000.** The 802 Spike positive samples in our assay were grouped in bins of 0.1 OD units with the initial bin starting at 0.4039, the minimum positive Spike OD value. The maximum Spike OD of all samples was 2.998.

**Table 2. Comparison of S protein ELISA versus Luminex immunodetection of S, RBD, N protein and status of the 4 hCoVs and MERS and SARS-CoV-1.**

| | ELISA | | Luminex | | | | | | | | |
|---|---|---|---|---|---|---|---|---|---|---|---|
| Negative controls (sample numbers) | Spike protein OD | RBD protein OD | SARS-CoV-2 Spike | SARS-CoV-2 Spike RBD | SARS-CoV-2 Nucleocapsid Protein | hCoV-229E Spike S1 | hCoV-HKU1 Spike S1 | hCoV-NL63 Spike S1 | hCoV-OC43 Spike S1 | MERS-CoV Spike S1 | SARS-CoV Spike S1 |
| N1 | 0.132 | 0.312 | 16.5 | 25.5 | 28.5 | 752.3 | 427.5 | 681.5 | 1451.5 | 18.5 | 25.5 |
| N2 | 0.159 | 0.196 | 39 | 4.5 | 2 | 1480.8 | 78.5 | 352.5 | 287.5 | 1.5 | 17.5 |
| N3 | 0.138 | 0.136 | 19 | 10.5 | 11 | 3016.3 | 539.5 | 538 | 1343.5 | 5.5 | 16.5 |
| N4 | 0.127 | 0.11 | 49 | 26.5 | 1091.5 | 844.3 | 1322.5 | 90 | 170.5 | 7.5 | 22.5 |
| N5 | 0.056 | 0.018 | 30.5 | 3.5 | 30.5 | 726.3 | 525.5 | 462.5 | 543.5 | 1.5 | 6.5 |
| N6 | 0.251 | 0.21 | 29 | 57 | 73.5 | 3204.8 | 257.5 | 433.5 | 581 | 43.5 | 45 |
| N7 | 0.178 | 0.178 | 17 | 41 | 45.5 | 1494.8 | 253.5 | 777.5 | 213 | 31.5 | 33.5 |
| N8 | 0.203 | 0.181 | 113 | 38 | 17.5 | 613.3 | 835.5 | 287.5 | 545.5 | 10 | 13.5 |
| N9 | 0.128 | 0.12 | 74.5 | 12.5 | 15 | 1419.8 | 696 | 412 | 293.5 | 3.5 | 12.5 |
| N10 | 0.152 | 0.207 | 61 | 25.5 | 19.5 | 2794.8 | 456.5 | 938.5 | 506 | 7.5 | 14.5 |
| N11 | 0.089 | 0.185 | 14 | 14.5 | 8.5 | 1496.8 | 548.5 | 463.5 | 596.5 | 3.5 | 10.5 |
| | ELISA | | Luminex | | | | | | | | |
| Positive Spike and Positive RBD Samples | Spike protein OD | RBD protein OD | SARS-CoV-2 Spike | SARS-CoV-2 Spike RBD | SARS-CoV-2 Nucleo-capsid Protein | hCoV-229E Spike S1 | hCoV-HKU1 Spike S1 | hCoV-NL63 Spike S1 | hCoV-OC43 Spike S1 | MERS-CoV Spike S1 | SARS-CoV Spike S1 |
| B1 | 2.904 | 2.991 | 21290.3 | 8917.7 | 5821.3 | 260.5 | 320.3 | 768.3 | 81.7 | 21.7 | 128.3 |
| B3 | 2.111 | 2.701 | 14502.8 | 4787.2 | 4541.3 | 3033 | 1447.3 | 403.3 | 1511.2 | 7.7 | 138.3 |
| B4 | 2.637 | 3.020 | 24563.3 | 11541 | 7229.3 | 3173 | 1588.3 | 477.3 | 583.2 | 11.7 | 100.3 |
| B5 | 3.000 | 2.944 | 16272.3 | 6672.2 | 4689.3 | 2183 | 1399.3 | 593.8 | 360.2 | 7.7 | 44.8 |
| B6 | 2.218 | 3.125 | 25727.3 | 13068 | 6875.8 | 2881.5 | 287.3 | 1614.3 | 755.7 | 8.7 | 61.3 |
| B7 | 1.380 | 3.049 | 24344 | 8703 | 7509 | 3266 | 970 | 1208 | 1324.5 | 11 | 106 |
| B8 | 1.432 | 3.132 | 25231.3 | 10711 | 6537.3 | 856 | 821.3 | 626.3 | 1900.2 | 23.7 | 103.3 |
| B9 | 2.019 | 2.952 | 16536.8 | 6820.7 | 5045.3 | 4267.5 | 460.8 | 2000.3 | 1277.7 | 10.2 | 153.3 |
| B10 | 2.643 | 3.042 | 27812.8 | 17379 | 9306.3 | 1560 | 1060.3 | 1241.3 | 406.2 | 9.7 | 403.3 |
| B11 | 2.701 | 2.662 | 23273.8 | 9145.7 | 14334.3 | 3689 | 332.3 | 1295.8 | 598.7 | 21.7 | 259.3 |
| | ELISA | | Luminex | | | | | | | | |
| Sample number <1.0 for Spike, RBD neg | Spike protein OD | RBD protein OD | SARS-CoV-2 Spike | SARS-CoV-2 Spike RBD | SARS-CoV-2 Nucleocapsid Protein | hCoV-229E Spike S1 | hCoV-HKU1 Spike S1 | hCoV-NL63 Spike S1 | hCoV-OC43 Spike S1 | MERS-CoV Spike S1 | SARS-CoV1 Spike S1 |
| C1 | 0.506 | 0.159 | 35 | 47 | 78 | 1431 | 3386 | 593 | 2136 | 4 | 99 |
| C2 | 0.508 | 0.097 | 352 | 13 | 40 | 5246 | 1581 | 342 | 2480 | 3 | 13 |
| C3 | 0.51 | 0.133 | 47 | 41 | 203 | 438 | 3263 | 183 | 3974 | 4 | 84 |
| C4 | 0.51 | 0.258 | 300 | 13 | 24 | 505 | 405 | 82 | 838 | 2 | 15 |
| C5 | 0.513 | 0.169 | 431 | 15 | 48 | 1801 | 1455 | 677 | 4787 | 2 | 20 |
| C6 | 0.516 | 0.134 | 243 | 11 | 12 | 2062 | 797 | 138 | 1793 | 2 | 13 |
| C7 | 0.553 | 0.104 | 430 | 32 | 470 | 947 | 443 | 356 | 1160 | 13 | 28 |
| C8 | 0.561 | 0.065 | 1297 | 16 | 21 | 776 | 1873 | 741 | 2410 | 4 | 20 |
| C9 | 0.563 | 0.146 | 578 | 20 | 34 | 5654 | 3184 | 1058 | 5784 | 6 | 22 |
| C10 | 0.566 | 0.234 | 450 | 13 | 30 | 3929 | 597 | 289 | 1405 | 8 | 11 |
| C11 | 0.571 | 0.043 | 343 | 18 | 32 | 4008 | 1637 | 698 | 4811 | 3 | 19 |
| C12 | 0.574 | 0.222 | 1086 | 11 | 24 | 1004 | 1753 | 816 | 2718 | 4 | 16 |
| C13 | 0.579 | 0.298 | 44 | 25 | 60 | 1126 | 796 | 568 | 1911 | 2 | 14 |
| C14 | 0.581 | 0.191 | 580 | 7 | 41 | 659 | 1517 | 351 | 2412 | 2 | 64 |

(Continued)

**Table 2.** (Continued)

| | | | | | | | | | | | |
|---|---|---|---|---|---|---|---|---|---|---|---|
| C15 | 0.587 | 0.255 | 213 | 11 | 18 | 3960 | 3817 | 667 | 3054 | 2 | 3 |
| C16 | 0.597 | 0.313 | 59 | 20 | 42 | 1292 | 1934 | 319 | 1182 | 5 | 21 |
| C17 | 0.601 | 0.309 | 2517 | 423 | 363 | 267 | 1089 | 73 | 2073 | 4 | 88 |
| c20 | 0.602 | 0.284 | 36 | 83 | 95 | 1954 | 3003 | 258 | 4087 | 6 | 57 |
| C21 | 0.613 | 0.319 | 25 | 24 | 51 | 3276 | 2953 | 206 | 547 | 3 | 18 |
| C22 | 0.614 | 0.173 | 413 | 35 | 49 | 3405 | 2164 | 551 | 2615 | 4 | 44 |
| C23 | 0.615 | 0.225 | 17 | 6 | 22 | 3331 | 206 | 406 | 1701 | 3 | 12 |
| C24 | 0.621 | 0.173 | 2352 | 1552 | 5752 | 218 | 1205 | 66 | 3895 | 8 | 124 |
| C25 | 0.629 | 0.342 | 25 | 12 | 27 | 1690 | 2301 | 640 | 149 | 4 | 22 |
| C26 | 0.636 | 0.08 | 276 | 132 | 249 | 750 | 636 | 304 | 639 | 1 | 179 |
| C27 | 0.639 | 0.337 | 582 | 190 | 502 | 4874 | 3450 | 240 | 3941 | 5 | 389 |
| C28 | 0.641 | 0.055 | 29 | 7 | 22 | 824 | 313 | 296 | 157 | 3 | 11 |
| C29 | 0.642 | 0.15 | 223 | 24 | 68 | 2828 | 2071 | 376 | 1308 | 2 | 16 |
| C30 | 0.645 | 0.22 | 156 | 102 | 240 | 3734 | 1090 | 245 | 2153 | 11 | 129 |
| C31 | 0.671 | 0.299 | 7 | 11 | 27 | 932 | 352 | 332 | 351 | 3 | 26 |
| C32 | 0.675 | 0.149 | 144 | 44 | 62 | 2791 | 650 | 384 | 506 | 5 | 51 |
| C33 | 0.684 | 0.399 | 100 | 20 | 32 | 392 | 306 | 206 | 645 | 3 | 20 |
| C34 | 0.695 | 0.332 | 851 | 13 | 31 | 2196 | 615 | 848 | 210 | 6 | 14 |
| C35 | 0.696 | 0.246 | 10 | 10 | 12 | 871 | 201 | 290 | 485 | 2 | 3 |
| C36 | 0.701 | 0.373 | 27 | 14 | 60 | 2098 | 833 | 244 | 2285 | 2 | 16 |
| C37 | 0.706 | 0.133 | 53 | 71 | 37 | 1322 | 489 | 302 | 1443 | 3 | 49 |
| C38 | 0.713 | 0.094 | 1796 | 9 | 18 | 5052 | 347 | 879 | 1168 | 4 | 14 |
| C39 | 0.72 | 0.304 | 1676 | 10 | 12 | 1118 | 1872 | 36 | 3155 | 3 | 11 |
| C40 | 0.726 | 0.348 | 389 | 41 | 33 | 1842 | 4402 | 598 | 2966 | 5 | 43 |
| C41 | 0.727 | 0.309 | 70 | 13 | 9 | 1223 | 548 | 241 | 3071 | 2 | 10 |
| C42 | 0.749 | 0.377 | 891 | 480 | 585 | 1891 | 1783 | 91 | 2138 | 6 | 344 |
| C43 | 0.765 | 0.19 | 2001 | 12 | 24 | 1519 | 259 | 473 | 3312 | 2 | 11 |
| C44 | 0.771 | 0.13 | 13 | 11 | 85 | 2403 | 1090 | 598 | 1536 | 3 | 16 |
| C45 | 0.781 | 0.131 | 1096 | 26 | 466 | 3722 | 1554 | 1781 | 1731 | 10 | 21 |
| C46 | 0.785 | 0.084 | 790 | 152 | 287 | 4108 | 5443 | 171 | 3325 | 4 | 258 |
| C47 | 0.791 | 0.205 | 330 | 11 | 18 | 3357 | 1197 | 183 | 1453 | 1 | 15 |
| C48 | 0.809 | 0.082 | 1546 | 648 | 33 | 1676 | 622 | 439 | 754 | 5 | 9 |
| C49 | 0.809 | 0.141 | 1478 | 80 | 166 | 3797 | 2331 | 960 | 1709 | 11 | 124 |
| C50 | 0.813 | 0.189 | 2329 | 180 | 459 | 307 | 1825 | 384 | 1959 | 3 | 335 |
| C51 | 0.815 | 0.372 | 140 | 39 | 104 | 1083 | 219 | 550 | 217 | 3 | 49 |
| C52 | 0.861 | 0.32 | 233 | 30 | 119 | 425 | 741 | 125 | 2896 | 9 | 30 |
| C53 | 0.883 | 0.122 | 19 | 45 | 41 | 214 | 204 | 187 | 623 | 2 | 48 |
| C54 | 0.898 | 0.161 | 701 | 10 | 19 | 2354 | 275 | 100 | 2070 | 5 | 20 |
| C55 | 0.961 | 0.321 | 979 | 75 | 144 | 2244 | 3249 | 1127 | 2419 | 2 | 65 |
| C56 | 0.976 | 0.194 | 2897 | 41 | 84 | 4761 | 2614 | 798 | 807 | 42 | 63 |
| C57 | 0.996 | 0.097 | 3252 | 837 | 353 | 816 | 731 | 244 | 1570 | 4 | 20 |

Samples consisted of 11 negative controls from the S protein ELISA, 11 that were high positive for both S and RBD protein in the ELISA, and 57 samples with a progressive increase in S protein OD value (2nd column) from the ELISA, but that were RBD negative. The Luminex threshold for positivity was set from the negative controls as the average plus 3 standard deviations for SARS-CoV-2 S protein = 134.05, RBD protein of = 73.63, and Nucleocapsid protein = 1088.50. Two columns showing data from the S protein and RBD protein ELISA are given for comparison to all the Luminex immunodetection data for S protein, RBD, Nucleocapsid, hCoV-229E S1, hCoV-HKU1 S1, hCoV-NL63 S1, hCoV-OC43 S1, MERS-CoV S1 and SARS-CoV-1 S1. No correlation was found between SARS-CoV-2 Spike and hCoV-229E Spike S1 (-0.275), hCoV-HKU1 Spike S1 (-0.219), hCoV-NL63 Spike S1 (0.159), hCoV-OC43 Spike S1 (-0.138), MERS-CoV Spike S1 (0.310) or SARS-CoV Spike S (0.359). Samples selected as "highly positive" for Spike and RBD levels were randomly selected samples at least 10 standard deviations greater than average negative control Spike OD value. For positive Spike and positive RBD samples, p values of Spike OD to RBD (<0.00001) and to Nucleocapsid protein (p<0.000001) were significant. The green boxes are values that were considered positive in the Luminex assay.

**Table 3. Assessment of raw S protein OD values in donors with 2 or more donations analyzed over 2 time periods from August 13th to December 8th of 2020.**

|  | N | Mean Pos OD | Mean # of days | p value |
|---|---|---|---|---|
| Pos-Neg | 38 | 0.596 | 52.8 |  |
| Stayed Pos | 24 | 1.466 | 69.3 | 0.0119 |
| All Positives | 802 | 1.228 |  |  |
|  | Number of Spike OD positive | N | % positive | p value |
| All | 802 | 9550 | 8.40 |  |
| 1st 58 days | 362 | 4786 | 7.56 |  |
| 2nd 58 days | 440 | 4764 | 9.24 | 0.0016 |

The top part of the table shows the Pos-Neg donors that were positive for S protein antibodies on first read only versus the "Stayed-Pos" donors that had at least 2 positive S protein antibody readings over the time shown in days. The bottom part of the table shows the first 58 days spanned from August 13th to October 10th, 2020 versus the second 58-day period spanning from the rest of October 11th to December 8th, 2020.

correlation between antibodies against the S1 protein from these 4 hCoVs and positivity or lack of positivity for SARS-CoV-2 S or RBD proteins (Table 2). These results support the validity of the ELISA but also suggest that the ELISA is more sensitive than the Luminex platform in detecting S protein of SARS-CoV-2.

Of the 802 S protein ELISA positive donors, 108 donated at least 2 times from August 13th to December 8th, 2020, and hence the study monitored maintenance or loss of antibody reactivity over time (Table 3). Analysis showed 38 donors had significant antibodies against S protein on their first donation but not on the second donation, with an average time of 53 days (Table 3). In contrast, 24 donors maintained significant antibody reactivity and remained positive between the 2 donations, with an average time span of 69 days. However, of the 38 that lost positivity by the 2nd donation the initial composite ELISA OD value was 0.596, while the group of 24 donors that maintained ELISA positivity had a much higher OD value of 1.47 (Table 3). Thus, the group of 38 repeat donors that lost their antibody positive status on the 2nd donation likely reflects the low starting point of antibody reactivity in conjunction with the known gradual loss of antibodies to SARS-CoV-2 over time [16], which now was below the sensitivity of the assay. It was also interesting that younger donors were more likely to lose their S protein ELISA positivity between their first and second donations, while those over 51 years of age were more likely to maintain S protein reactivity (Table 4).

Sample collection from August 13th—December 8th of 2020 was split in half, which showed a significant increase in rates of positivity over time (Table 3). Specifically, rates of S protein ELISA positivity were 7.56% across the GCMA from August 13th through October 10th (58 days), compared with 9.24% from October 10th through December 8th, 2020 (58 days)

**Table 4. Comparison of age of donors losing S protein positivity versus donors maintaining positivity.**

| Age group | Total | Pos-Neg | Pos-Pos |
|---|---|---|---|
| 16–30 | 7 | 7 | 0 |
| 31–40 | 3 | 3 | 0 |
| 41–50 | 5 | 4 | 1 |
| 51–60 | 16 | 9 | 7 |
| 60+ | 31 | 15 | 16 |
| Total | 62 | 38 | 24 |

Fourteen of 15 donors 50 and under lost antibody titer compared to 24 of 47 donors over 51.

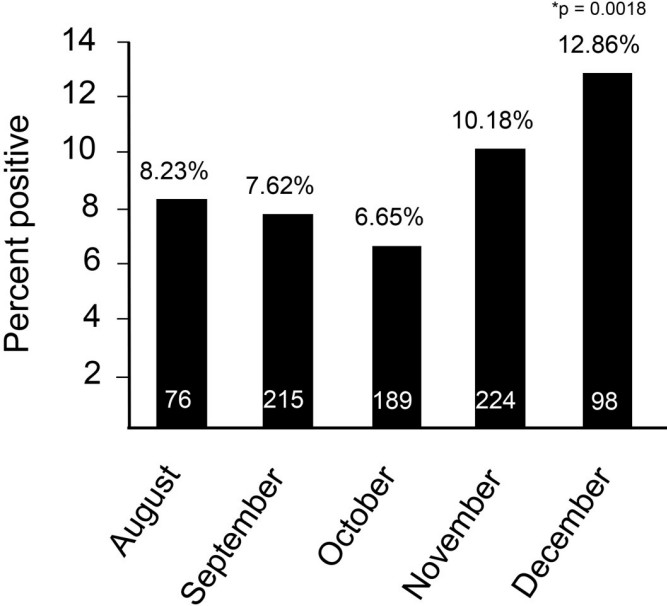

**Fig 2. Percent positive S protein ELISA seroprevalence by month from August 13, 2020 to December 8, 2020.**
Monthly breakdown of the percent positive rate for 9550 unique donors. In August 76 of 924 donors were positive (8.225%), September 215 of 2821 donors were positive (7.621%), October 189 of 2842 donors were positive (6.650%), November 224 of 2201 donors were positive (10.177%), and December 98 of 762 donors were positive (12.861%). *p-value was determined from the z score for each month compared to the August positive rate for significance. Only December was significant as compared to August (z = -3.114, p = 0.0018 Two-tailed).

(Table 3). Rates of S protein reactivity were also analyzed by month, which showed a temporal increase, culminating in a value of 12.86% in the portion of December that was evaluated (Fig 2). Thus, the GCMA emerged into the national vaccination phase of the pandemic with a background level of ≥13% of individuals with some level of immunity against SARS-CoV-2 (this level is likely much higher, see Discussion).

Within the donor dataset, age ranges were also evaluated for rates of prior infection. The 9550 donors spanned in age from 16 to 91 years, which was broken into roughly decade increments. Interestingly, the donors from 16–30 years of age had the highest rates of antibodies against S protein compared with individuals in their 30s, 40s, and 50s while adults over 60 years of age had significantly lower rates (Table 5). Finally, gender association was 8.52% in males versus 8.28% in females, which was not statistically different (Table 5).

**Table 5. Rates of S protein antibody positivity by ELISA from the indicated donor age ranges or by gender.**

|  |  | Number of donors | Number of Spike OD positive | Percent positive | p value |
|---|---|---|---|---|---|
| Age group of donors | 16–30 | 1442 | 136 | 9.43 | 0.0286 |
|  | 31–40 | 1447 | 108 | 7.46 | n.s. |
|  | 41–50 | 1571 | 140 | 8.91 | n.s. |
|  | 51–60 | 2185 | 200 | 9.15 | n.s. |
|  | 60+ | 2905 | 218 | 6.98 | 0.0144 |
| Sex of donors | Male | 4564 | 389 | 8.52 |  |
|  | Female | 4986 | 413 | 8.28 | n.s. |

Only the 16–30 age range and the 60+ age range was statistically different from 31–40 age range, but the 31–40, 41–50 and 51–60 were not statistically different from each other. Rates in males and females were not significantly different (n.s.).

**Table 6. Rate of S protein ELISA positivity clustered by regions within the GCMA.**

| Ohio* | Number of donors | Number of positive donors | Percent positive | p value |
|---|---|---|---|---|
| West of I-75 | 3136 | 302 | 9.63 | |
| East of I-75 | 4266 | 347 | 8.14 | 0.0123 |
| Ohio | 7418 | 652 | 8.79 | |
| Kentucky | 2132 | 150 | 7.04 | 0.005 |

*Excludes Kentucky and 17 people (3 positive) in zip codes divided evenly by I-75. However, Ohio was also summated along with the additional 17 donors on the I-75 border to compare against Kentucky zip codes on the south side of the Ohio River, but within the GCMA.

We also analyzed geographic subregions within the GCMA for rates of S protein antibodies. Unfortunately, sampling was not large enough to examine rates based on individual zip codes, although statistical evaluation of the GCMA as larger subregions was possible, such as West versus East side, as split by Interstate 75 (I-75), and as rates in Ohio versus Kentucky, as split by the Ohio River (Table 6 and Fig 2). The data show a rate of 9.63% on the West side of Cincinnati versus 8.13% on the East side, while the Ohio portion of the GCMA was 8.79% versus 7.03% in the adjoining Kentucky region (Table 6 and Fig 3).

## Discussion

To our knowledge the current study is the first to report rates of SARS-CoV-2 seroprevalence in the GCMA immediately preceding the national vaccination program. The total cumulative rate as of December 2020 was ~13%, and this background rate of past infectivity will factor into the goal of achieving herd immunity in conjunction with the national vaccination program [17,18]. Indeed, tracking of 150,000 previously infected individuals in Ohio and Florida from March 2020 to August 2020 showed that these individuals were relatively protected from re-infection, like vaccination [6]. In the current study the highest rates of past infection were observed in donors under 30 years of age, and more generally on the West side of the GCMA compared to the East side and adjoining regions of Kentucky. Data trends failed to reveal a difference in background levels of past infectivity based on race in the GCMA as analyzed in White, non-Hispanics, African Americans, Hispanics or Latinos, and Asians, although the total sampling pool of the later 3 ethnic groups was too low for statistical certainty.

The detection by seroprevalence of past infection of SARS-CoV-2 is dependent on the quantitative measures of the S protein-based IgG-dependent ELISA. A modified ELISA protocol from Mount Sinai Icahn School of Medicine in New York City [9,10] was implemented, which was given an Emergency Use Authorization (EUA) by the U.S. FDA in April of 2020 [19]. Additional quality control measures included the use of 60 serum samples obtained prior to the onset of the pandemic as true negatives in generating a background value for the S protein ELISA. However, it is likely that the ELISA testing platform misses individuals in whom the levels of antibody dropped below assay detection, as previously reported [20]. Thus, not all individuals infected in the first half of the pandemic will maintain sufficient antibodies for the ELISA platform to detect, thus true rates of past infectivity in the GCMA are likely several percentage points higher (see below). The RBD domain is 85% smaller than the Spike protein and for that reason it is a less sensitive indicator of past infectivity, but analysis of RBD positivity still served as confirmation for the S protein-based ELISA and antibodies against the RBD region are more likely to be neutralizing against the virus.

Individuals who present to donate blood or related blood products are not a true cross-sectional representation of a metropolitan area. Indeed, donors are pre-screened for communicable diseases or behaviors that are high risk for attaining such diseases. Moreover, the ethnicity

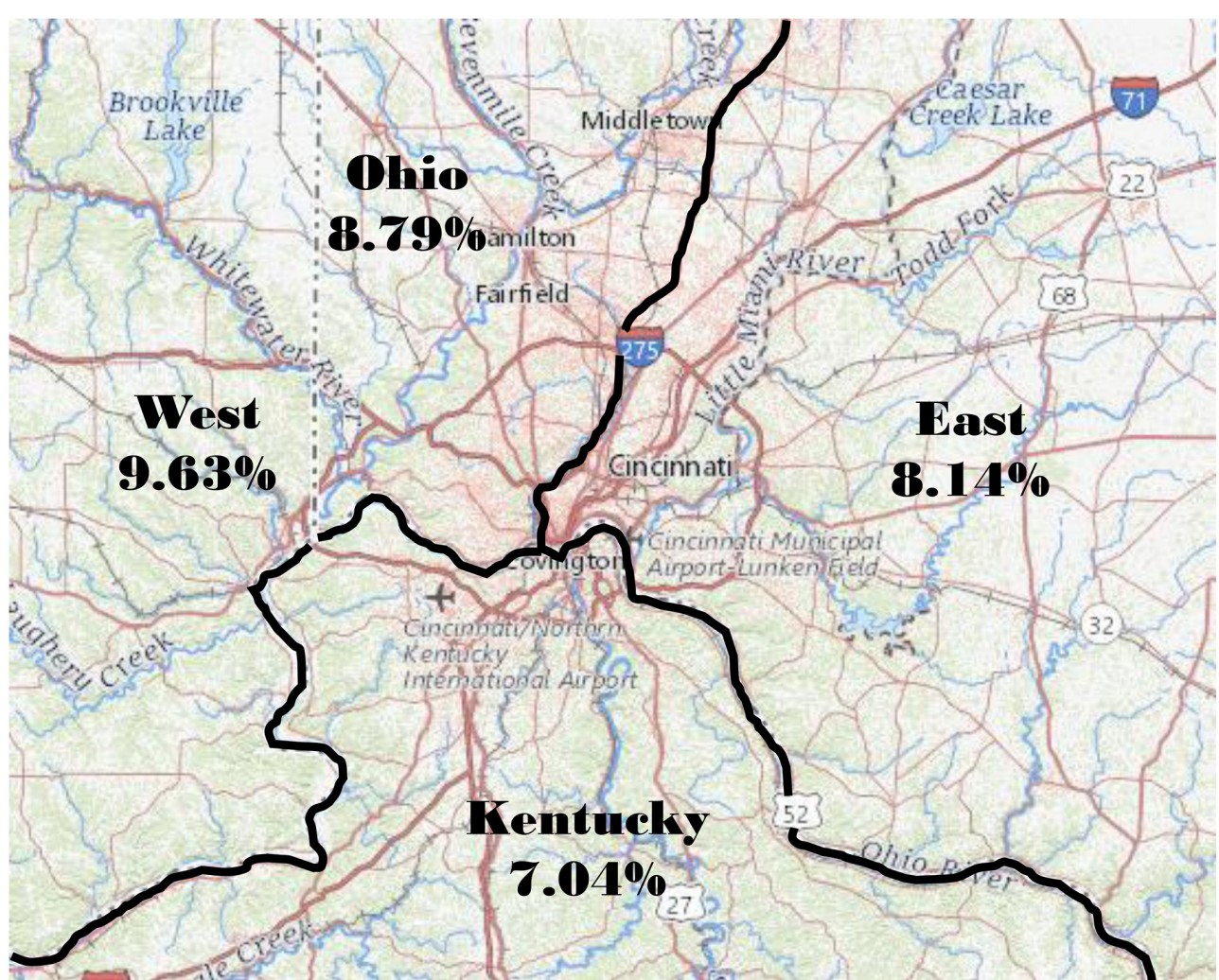

**Fig 3. Seroprevalence of Spike protein ELISA positivity in the Greater Cincinnati Metropolitan Area.** Map image courtesy of the Unites States Geological Survey (USGS) National Map Viewer (public domain: https://apps.nationalmap.gov/viewer/). The Greater Cincinnati Metropolitan Area (GCMA) shown as defined by the United States Census Bureau (https://data.census.gov/cedsci/map?q=All%20counties%20in%20Ohio&g= 310M500US17140&tid=ACSDP5Y2019.DP02&layer=VT_2019_310_M5_PY_D1&cid=DP02_0093PE&palette=Teal&break=5&classification=Natural %20Breaks&mode=customize) and reported in the 2010 AGE, RACIAL, GENDER AND MARITAL STRUCTURE OF GREATER CINCINNATI. The Cincinnati Metropolitan Statistical Area (MSA) includes: Butler, Brown, Clermont, Hamilton, and Warren Counties in Ohio; Boone, Bracken, Campbell, Gallatin, Grant, Kenton, and Pendleton Counties in Kentucky; and Dearborn, Franklin, and Ohio Counties in Indiana.

of the GCMA blood donor volunteers differs from the national race composition within the USA. In our dataset 88% of individuals presenting to a blood center in the GMCA were White, non-Hispanic, 3.3% Hispanic, 4.7% African American, and 2.3% Asian, so White, non-Hispanics are over-represented in our data set, which is comparable to race distribution of blood donors observed at other centers across the USA [21–23]. According to current centers for disease control (CDC) data, the White, non-Hispanic population accounts for 50% of COVID cases, Hispanic/Latino 28.9%, African Americans 11.2%, and Asian 3.2%. However, we cannot make direct inferences about the overall prevalence of SARS-CoV-2 infectivity rates in the GCMA based on race given the low sampling size of Hispanic, African American, and Asians in our dataset. Also, our data set was dramatically over-represented by White, non-Hispanic donors as previously observed with blood donation [21–23]. Finally, blood donors tend to be

healthy and have lower prevalence of acquired chronic health conditions. However, given that our current study was a post-hoc analysis of de-identified specimens, no selection bias based on donor suspicions of past infection was involved.

In the past year numerous seroprevalence studies have been published or uploaded to pre-print servers from across the USA, although few have extended to December 2020. A few of these past studies are relevant and interesting to consider in relation to our current analysis. One such study examined 252,882 blood donors over 24 centers across the USA from the months of June and July 2020 [22]. Vassallo et al., utilized the Ortho VITROS Anti-SARS-CoV-2 total immunoglobulin assay for the S1 region of the S protein for IgG, IgA and IgM [22], which was different from the full-length S protein ELISA detecting IgG serum levels that was produced in house and employed here. Vassallo et al., reported a rate of 1.83% in June and 2.26% in July across their USA sample population. Within these data were results from Chicago, which showed a rate of 2.76% in June and 3.34% in July [22]. By comparison, another seroprevalence study from the Chicago area that analyzed 1545 solicited volunteers, showed a seroprevalence of 19.8% from June 24 through September 6, 2020 [24]. It seems likely that select community-based recruitment variables and the technical aspects of the immunodetection platform underlies the widely disparate results discussed here. However, based on PCR measured SARS-CoV-2 molecular detection from the beginning of the pandemic until March of 2021, there were approximately 985,000 Ohioans infected with the virus. This equates to a rate of 8.27% of the state's population, which is generally consistent our data showing a rate of ~13% seropositivity as of December of 2020 in the GCMA. More interestingly, extrapolation of these data 3 additional months to March 2021, the rate of past infectivity in the GCMA becomes ~16–17%, and as stated above this approximation under-estimates the true rate due to the known gradual decline in antibody levels below the threshold of the S protein-based ELISA [16,20]. Given all these factors an overall estimated rate of past infection within the GCMA as of March 2021 is ~20%.

Another study examined 177,919 seemingly random adult blood samples from across 50 States that spanned from July 27 through September 24, reporting a range of just under 1% to over 20%. This study used 3 different automated clinical laboratory immunodetection platforms for either S or N protein. Their data show rates within Ohio of 2.8 to 5.0% for donor samples collected from August 24—September 24, 2020, which overlaps with part of our collection time [25]. This value is consistent with another more limited seroprevalence study with 727 samples from across the entire state of Ohio in July 2020, which also incorporated mathematical modeling to predict a rate of 7% past viral infectivity [26], a value that is close to our data from the adjacent month of August in the GCMA.

The past studies discussed above generally support our conclusions and suggest that the ELISA implemented here was rigorous and properly calibrated. The data in this study establish a rate of ~13% past SARS-CoV-2 infectivity within the GMCA by the end of 2020, and extrapolation to the present day (March of 2021) approximates a rate of 16–17% past infectivity, and likely even >20% if depreciation in blood antibody levels over time is considered [20]. This knowledge can impact the deployment of the vaccination program towards more rapidly achieving herd immunity [18]. For example, previously infected individuals might only require 1 vaccine dosage for full protection compared with a naïve individual who requires 2 vaccinations (with Moderna and Pfizer vaccines). Indeed, previously infected and recovered individuals produce a strong immunologic reaction after a single dosage of the Pfizer vaccine that is comparable to the standard 2 dose routine in naïve individuals [27]; and using this information and associated strategy would augment the relative supply of the vaccine in attempting to achieve herd immunity more rapidly.

## Supporting information

**S1 Fig. Expression, purification, and characterization of SARS-CoV-2 S and RBD proteins.** A) Comparison of expression yields in different cell lines. Dashed lines represent the reported yields from Stadlbauer et al. [10]. B) Reducing SDS-PAGE gel showing RBD purified by Ni-NTA and gel filtration chromatography and Spike protein purified by Ni-NTA chromatography. The full gel image is shown in this panel so there is no additional need to provide a separate Supporting Information file with this exact same full gel image. C-E) Sedimentation velocity analytical ultracentrifugation analysis of protein quality and assembly state in solution. C) Sedimentation coefficient distribution for RBD purified by Ni-NTA revealing monomer and disulfide-linked dimer species. D) Sedimentation coefficient distribution for monomeric RBD (experimental MW of 31.1 kDa) purified by S75 gel filtration. E) Sedimentation coefficient distribution for Spike protein showing that trimer is the predominant species (experimental MW of ~519 kDa). The trimer sedimentation coefficient of 12.9 S was consistent with the value (12.6 S) calculated by HullRad [28] hydrodynamic modeling of the glycosylated spike trimer structure (PDB 6VXX). The S protein monomer is predicted to sediment at approximately 5.5 S, but the exact value will depend on the hydrodynamic shape of the isolated monomeric species.
(TIF)

**S1 Data. De-identified raw data.** Excel spread sheet of the 9550 blood donors that were evaluated in this study broken into columns that shows the date of visit to the blood collection center, the State, the geographic region as east (E), west (W) or Kentucky (KY), the blood type, the age, gender, race and raw S protein ELISA OD value.
(XLSX)

## Acknowledgments

We would like to thank Dr Florian Krammer at Mount Sinai Icahn School of Medicine for providing the mammalian expression vectors for the modified S protein and the RBD protein [9,10], which were produced under HHSN272201400008C and obtained through BEI Resources, NIAID, NIH: Spike Glycoprotein Receptor Binding Domain (RBD) from SARS-Related Coronavirus 2, Wuhan-Hu-1 with C-Terminal Histidine Tag, Recombinant from HEK293F Cells, NR-52366 and Spike Glycoprotein (Stabilized) from SARS-Related Coronavirus 2, Wuhan-Hu-1 with C-Terminal Histidine Tag, Recombinant from Baculovirus, NR-52308.

## Author Contributions

**Conceptualization:** Monica Malone McNeal, Judith Gonzalez, E. Steve Woodle, Kristen Safier, Kristine A. Justus, Paul Spearman, Russell E. Ware, Jose A. Cancelas, Michael B. Jordan, Andrew B. Herr, David A. Hildeman, Jeffery D. Molkentin.

**Data curation:** Greg Davis, Willis Clark Bacon, Suh-Chin Lin, Alexander E. Yarawsky.

**Formal analysis:** Allen J. York, Judith Gonzalez, Jeffery D. Molkentin.

**Investigation:** Greg Davis, Willis Clark Bacon, Suh-Chin Lin, Joseph J. Maciag, Jeanette L. C. Miller, Kathryn C. S. Locker, Michelle Bailey, Rebecca Stone, Alyssa Sproles.

**Methodology:** Monica Malone McNeal, Alexander E. Yarawsky, Joseph J. Maciag, Jeanette L. C. Miller, Kathryn C. S. Locker, Michelle Bailey, Rebecca Stone, Michael Hall, Judith Gonzalez, Alyssa Sproles.

**Project administration:** Jeffery D. Molkentin.

**Resources:** Rebecca Stone, Michael Hall, Judith Gonzalez, E. Steve Woodle, Kristen Safier, Kristine A. Justus, Jose A. Cancelas.

**Supervision:** Paul Spearman, Russell E. Ware, Jose A. Cancelas, Michael B. Jordan, Andrew B. Herr, David A. Hildeman, Jeffery D. Molkentin.

**Writing – original draft:** Jeffery D. Molkentin.

**Writing – review & editing:** Jeffery D. Molkentin.

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
