## [Decision Letter · Decision Letter 0]

17 May 2021

PONE-D-21-09100

Seroprevalence of SARS-CoV-2 Infection in Cincinnati Ohio USA from August to December 2020

PLOS ONE

Dear Dr. Molkentin,

Thank you for submitting your manuscript to PLOS ONE. After careful consideration, we feel that it has merit but does not fully meet PLOS ONE’s publication criteria as it currently stands. Therefore, we invite you to submit a revised version of the manuscript that addresses the points raised during the review process.

We look forward to receiving your revised manuscript.

Kind regards,

Binod Kumar, PhD

Academic Editor

PLOS ONE

Journal Requirements:

2. In your Methods section, please provide additional information about the biological samples and the demographic details of the donors. Please ensure you have provided sufficient details to replicate the analyses such as:

a) the date(s) researchers accessed the samples,

b) the date range that samples were originally collected,

c) a description of any inclusion/exclusion criteria that were applied to sample selection,

d) a table of relevant demographic details of donors (if disclosed), \\\\.

3. Please provide sequences or accession numbers for the SARS-CoV-2 antigens for S protein and RBD used in the study.

4. For studies involving humans categorized by race/ethnicity, age, disease/disabilities, religion, sex/gender, sexual orientation, or other socially constructed groupings, authors should:

1) Explicitly describe their methods of categorizing human populations,

2) Define categories in as much detail as the study protocol allows,

3) Update outmoded terms and potentially stigmatizing labels to more current, acceptable terminology.

Examples: “Caucasian” should be changed to “white”

5. Thank you for stating the following in the Financial Disclosure section:

"This study was supported the Howard Hughes Medical Institute (to J.D.M. No grant number, Molkentin Investigator). J.D.M was also supported by an internal grant from Cincinnati Children Research Foundation to conduct these studies on COVID-19 (Molkentin grant 1).   Contributions to this work by P.S. were financially assisted by The John Hauck Foundation, Fifth Third Bank.

https://www.cincinnatichildrens.org/

https://fconline.foundationcenter.org/fdo-grantmaker-profile?key=HAUC002

https://www.hhmi.org/

The funders had no role in study design, data collection and analysis, decision to publish, or preparation of the manuscript"

We note that you received funding from a commercial source: Fifth Third Bank.

6. We noted in your submission details that a portion of your manuscript may have been presented or published elsewhere. [https://www.medrxiv.org/content/10.1101/2021.03.11.21253263v1.supplementary-material] Please clarify whether this publication was peer-reviewed and formally published. If this work was previously peer-reviewed and published, in the cover letter please provide the reason that this work does not constitute dual publication and should be included in the current manuscript.

7. We note that you have indicated that data from this study are available upon request. PLOS only allows data to be available upon request if there are legal or ethical restrictions on sharing data publicly. For information on unacceptable data access restrictions, please see http://journals.plos.org/plosone/s/data-availability#loc-unacceptable-data-access-restrictions.

8. PLOS ONE now requires that authors provide the original uncropped and unadjusted images underlying all blot or gel results reported in a submission’s figures or Supporting Information files. This policy and the journal’s other requirements for blot/gel reporting and figure preparation are described in detail at https://journals.plos.org/plosone/s/figures#loc-blot-and-gel-reporting-requirements and https://journals.plos.org/plosone/s/figures#loc-preparing-figures-from-image-files. When you submit your revised manuscript, please ensure that your figures adhere fully to these guidelines and provide the original underlying images for all blot or gel data reported in your submission. See the following link for instructions on providing the original image data: https://journals.plos.org/plosone/s/figures#loc-original-images-for-blots-and-gels.

9. We note that you have included the phrase “data not shown” in your manuscript. Unfortunately, this does not meet our data sharing requirements. PLOS does not permit references to inaccessible data. We require that authors provide all relevant data within the paper, Supporting Information files, or in an acceptable, public repository. Please add a citation to support this phrase or upload the data that corresponds with these findings to a stable repository (such as Figshare or Dryad) and provide and URLs, DOIs, or accession numbers that may be used to access these data. Or, if the data are not a core part of the research being presented in your study, we ask that you remove the phrase that refers to these data.

10. Your ethics statement should only appear in the Methods section of your manuscript. If your ethics statement is written in any section besides the Methods, please move it to the Methods section and delete it from any other section. Please ensure that your ethics statement is included in your manuscript, as the ethics statement entered into the online submission form will not be published alongside your manuscript.

11. We note that Figure 2 in your submission contain map images which may be copyrighted. All PLOS content is published under the Creative Commons Attribution License (CC BY 4.0), which means that the manuscript, images, and Supporting Information files will be freely available online, and any third party is permitted to access, download, copy, distribute, and use these materials in any way, even commercially, with proper attribution. For these reasons, we cannot publish previously copyrighted maps or satellite images created using proprietary data, such as Google software (Google Maps, Street View, and Earth). For more information, see our copyright guidelines: http://journals.plos.org/plosone/s/licenses-and-copyright.

11.1.    You may seek permission from the original copyright holder of Figure 2 to publish the content specifically under the CC BY 4.0 license. 

11.2.    If you are unable to obtain permission from the original copyright holder to publish these figures under the CC BY 4.0 license or if the copyright holder’s requirements are incompatible with the CC BY 4.0 license, please either i) remove the figure or ii) supply a replacement figure that complies with the CC BY 4.0 license. Please check copyright information on all replacement figures and update the figure caption with source information. If applicable, please specify in the figure caption text when a figure is similar but not identical to the original image and is therefore for illustrative purposes only.

12. Please include captions for your Supporting Information files at the end of your manuscript, and update any in-text citations to match accordingly. Please see our Supporting Information guidelines for more information: http://journals.plos.org/plosone/s/supporting-information.

Reviewers' comments:

Reviewer's Responses to Questions

**Comments to the Author**

1. Is the manuscript technically sound, and do the data support the conclusions?

Reviewer #1: Yes

Reviewer #2: Yes

Reviewer #3: Yes

2. Has the statistical analysis been performed appropriately and rigorously? 

Reviewer #1: Yes

Reviewer #2: Yes

Reviewer #3: Yes

3. Have the authors made all data underlying the findings in their manuscript fully available?

Reviewer #1: Yes

Reviewer #2: Yes

Reviewer #3: Yes

4. Is the manuscript presented in an intelligible fashion and written in standard English?

Reviewer #1: Yes

Reviewer #2: Yes

Reviewer #3: Yes

5. Review Comments to the Author

Reviewer #1: The manuscript by Davis et al depicting the SAR-COv2 seroprevalence in Cincinnati Ohio USA for the time period August to December 2020 emphasizes on the rates of infection as well as serum IgG antibody for the viral Spike protein. ELISA seems more sensitive tool than Luminex for S-protein detection. They also monitored the average seropositive of S-antibody over the period of study to determine the loss of antibody. The age specific antibody production shows correlation between age groups and nullifies the fact of gender or ethnicity biasness. Their study showed that there is increase rate of infection in age group of below 30 as well as in later half of study which can be extrapolated to higher number for March 2021. This study also identified the geographical distribution of Cincinnati.

Overall, this study might help in National immunization program with a focus on development of herd immunity to the nation. I have few concerns that needs to be addressed:

1. Line 133/134, word “washed” missing.

2. The authors purified S and RBD domain ectopically expressing them in mammalian cell line. They have showed that 55.61% of S protein positive donors also positive for RBD domain. They did not clearly mention the reason of performing RBD positive donors. Most of the experiment that they performed were based on S protein only. This needs to be elaborated in discussion section.

3. Amongst the 108 donors who donated at least 2 times for this study showed 38 lost the antibody in second donation. It would be of more help for vaccination program to define the age group who lost S-protein antibody or whether it is some random phenomenon. Also same argument hold for 24 donors who showed positive for throughout the study.

4. The authors splitted for the study in two halves for better understanding of percent positivity of S-protein in 802 samples. What is the percentage positivity of Cov2 infection from samples collected for these two halves across Cincinnati or GCMA area?

5. The authors found some variation in S-protein positive cases as per geographic locations. Although the geographic variation is very little, a brief discussion might help readers to understand this variation within same state.

6. Line 314 - “March 2020” needs to be corrected to “March 2021”.

7. The National statistics by CDC shows that the highest rate of infection is almost 50% for White, Non-Hispanic followed by Hispanic population. The rate further drops to 11% for African- American and Asian had only3.3% of infectivity rate. How the authors can justify this variation in their samples in GCMA/Cincinnati area with respect to National infection rate?

8. Line 330 – Authors suggest that one vaccine dosage can give full protection with previously infected individuals. Given the data of loss of seropositivity in 38 donors of 108 for S-protein antibody on second test, authors needs to justify if this claim is safe overall by not giving two shots to them. Those individuals might not have the same level of antibody titer after having just one shot and might be of risk for potential infection instead of vaccination program which might put them in risk.

Reviewer #2: This study investigates the seroprevalence of SARS-CoV-2 infection amongst ~10K adult volunteers in the Cincinnati, USA by measuring serum IgG levels against the Spike protein (S). This is also compared to the serum IgG levels against the RBD protein. However, there seems to be no clear concordance between the two, which is alluded to the different sensitivity of the two ELISAs based on antigen size.

Please address the following comments:

1. Line 58 - The abstract should mention the exact timeline of the study, either 13th August to 8th December 2020, or mid August to first week of Dec, etc.

2. Line 99- please elaborate on how 'viral cellular involution' happens?

3. Line 112- it is not clear from this statement that the other HCoV strains were used to check cross-reactivity

4. Line 214- What was the OD cutoff for selecting the highly positive donors?

5. Line 216- What is the basis of 100% correlation? p-values would be useful here.

6. Line 308- reference?

7. Figure 1- Label X axis as OD units

8. Figure 2- Mention days/dates here. Or normalize the percent positivity by the no. of days for each month (especially for August and December)

9. Supplementary Figure 1E- Label the monomeric peak

10. Overall comment: what are the OD values in terms of antibody titers?

11. Were serum IgM levels against SARS-CoV-2 measured in these individuals?

Reviewer #3: Authors of the manuscript “Seroprevalence of SARS-CoV-2 Infection in Cincinnati Ohio USA from August to December 2020” have investigated rates of SARS-CoV-2 infection in 9,550 adult donors of the greater Cincinnati, Ohio, USA metropolitan area during the period of August to December 2020, just prior to initiation of the national vaccination program. They found overall positive prevalence of 8.40% for serum IgG antibody against the SARS-CoV-2 Spike protein. They also reported that males and females showed similar rates of past infection, and rates among different ethnic groups were not significantly different. They also reported lowest past infection rate among patients above the age of 60. Geographic analysis showed the lowest rates in the adjoining region of Kentucky (across the Ohio river). These findings about regional seroprevalence will help inform efforts to achieve herd immunity in conjunction with the national vaccination campaign. I would like to draw the attention of authors towards few points:

1. Line number 125/126 Board (IRB) reviewed the proposed SARS-CoV-2 serology initiative and classified it as non-human rearch. Please correct the sentence

2. The authors talked about significant variation in geographical analysis for past infection. Was comparison age matched as they observed age plays significant role so whether or not study population were matched for their age is important when considering geographical variations.

3. Authors should also discuss other studies with results showing variations in ethnic groups and should discuss their finding in comparison to CDC reports. They should discussion these studies too, if any done in the USA.

4. Detail statistical analysis for Table 4 will be helpful and a more explanatory table footnote will make it more clear to readers.

5. For figure 1.B p value must be indicated on the histogram

6. PLOS authors have the option to publish the peer review history of their article (what does this mean?). If published, this will include your full peer review and any attached files.

Reviewer #1: **Yes: **Rupkatha Mukhopadhyay

Reviewer #2: No

Reviewer #3: No

---

## [Author Response · Author response to Decision Letter 0]

7 Jun 2021

Reviewer #1: Overall, this study might help in National immunization program with a focus on development of herd immunity to the nation. I have few concerns that needs to be addressed:

1. Line 133/134, word “washed” missing.

Thank you, the word has been added, now lines 144/145.

2. The authors purified S and RBD domain ectopically expressing them in mammalian cell line. They have showed that 55.61% of S protein positive donors also positive for RBD domain. They did not clearly mention the reason of performing RBD positive donors. Most of the experiment that they performed were based on S protein only. This needs to be elaborated in discussion section.

We agree with the reviewer here and we have added a better rationale to both the results and discussion to address this concern. Lines 223-225 of the results present this issue and possible reasons for the discordance between S and RBD protein detection, which is also discussed now in lines 391-394 of the Discussion section. In fact, the bottom half of Table 1 directly addresses the sensitivity issue between S and RBD in the ELISA assay based on OD strength of the S protein results. 

3. Amongst the 108 donors who donated at least 2 times for this study showed 38 lost the antibody in second donation. It would be of more help for vaccination program to define the age group who lost S-protein antibody or whether it is some random phenomenon. Also same argument hold for 24 donors who showed positive for throughout the study.

To directly address this concern of the reviewer have analyzed our data to show age ranges and generated a new Table (Table 4). The data show that younger individuals showed loss of S protein ELISA positivity between the first and second blood donation, while individuals over 51 years of age were more likely to maintain their positive status (discussed in lines 304-306) 

4. The authors splitted for the study in two halves for better understanding of percent positivity of S-protein in 802 samples. What is the percentage positivity of Cov2 infection from samples collected for these two halves across Cincinnati or GCMA area?

Thank you for the comment and we have shown these data in the revised paper. Table 3 shows that of the 802 positive donors, the positivity rate was 7.56% for the 1st 58 days and 9.24% for the 2nd 58 days which we believe addresses this comment.

5. The authors found some variation in S-protein positive cases as per geographic locations. Although the geographic variation is very little, a brief discussion might help readers to understand this variation within same state.

We understand the reviewer's question here and why it would be interesting to speculate although we cannot conceive of a plausible reason why, for example, that the West side of Cincinnati had a higher rate of past infectivity compared with the East side, and Northern Kentucky had the lowest rate. Hence, we have elected to simple state the data and not attempt to interpret why there might sub-regional differences (For the reviewer's benefit, the "East side" of Cincinnati is typically more wealthy with higher property values while the "West side" has lower general property values and is more "rural" in many cultural characteristics). 

6. Line 314 - “March 2020” needs to be corrected to “March 2021”.

Thank you, this has been corrected. 

7. The National statistics by CDC shows that the highest rate of infection is almost 50% for White, Non-Hispanic followed by Hispanic population. The rate further drops to 11% for African- American and Asian had only 3.3% of infectivity rate. How the authors can justify this variation in their samples in GCMA/Cincinnati area with respect to National infection rate?

We thank the reviewer for this comment as it is an important issue and we have reformulated our discussion in lines 395-410 to discuss data from the CDC and past reports in the literature in comparison with our results related to race. Line 411-441 also deal more specifically with the total rate of past infectivity as detected by ELISA assays and why some of the differences in results may have occurred between select studies. 

8. Line 330 – Authors suggest that one vaccine dosage can give full protection with previously infected individuals. Given the data of loss of seropositivity in 38 donors of 108 for S-protein antibody on second test, authors needs to justify if this claim is safe overall by not giving two shots to them. Those individuals might not have the same level of antibody titer after having just one shot and might be of risk for potential infection instead of vaccination program which might put them in risk.

Our response to Comment 3 showed older individuals maintaining titer better, so potentially younger individuals could be targeted to ensure both doses are received, but older individuals might have a longer response and possibly would be more likely to sufficiently benefit with a single dose. We also bring up reference #27 in the discussion in lines 448-453 in speculating about 1 versus 2 vaccine dosages between naive and previously infected individuals, and what has already been shown to result in a strong immune response in each case.

Reviewer #2: 

1. Line 58 - The abstract should mention the exact timeline of the study, either 13th August to 8th December 2020, or mid August to first week of Dec, etc.

Thank you, this has been corrected to 13th August to 8th December 2020.

2. Line 99- please elaborate on how 'viral cellular involution' happens?

This reference to viral involution was part of our Introduction and we have modified this sentence and provided 2 references that discuss the basic biology of coronavirus involution through the S protein and the ACE2 receptor. We have also modified this sentence in the introduction, so it is more clear and points the reader to those reference for in-depth coverage of that topic (references 7,8) and lines 95-98.

3. Line 112- it is not clear from this statement that the other HCoV strains were used to check cross-reactivity using the 4 endemic human cold-causing coronaviruses (HCoV-229E, -NL63, -OC43, and -HKU1) (11,12) to measure baseline cross-reactivity.

This is an important issue and we have revised the interpretation of our data pertaining to the 4 endemic human cold-causing coronaviruses, in part because none of the Luminex data achieved statistical significance on their own, although our analysis of statistical correlation was informative and suggested that these readings were independent. More specifically, we selected a group of 11 highly positive donors for S and RBD proteins by ELISA for comparative analysis in the Luminex platform to associate with readings of the four coronaviruses which were non-correlative. However, this same platform showed 100% correlation between S protein ELISA values and Luminex values for S, RBD and Nucleocapsid (N) protein from SARS-CoV-2, but not S protein from MERS-CoV or SARS-CoV-1 (Table 2). 

Added to table legend: No correlation was found between SARS-CoV-2 Spike and hCoV-229E Spike S1 (-0.275), hCoV-HKU1 Spike S1 (-0.219), hCoV-NL63 Spike S1 (0.159), hCoV-OC43 Spike S1 (-0.138), MERS-CoV Spike S1 (0.310) or SARS-CoV Spike S (0.359).

4. Line 214- What was the OD cutoff for selecting the highly positive donors?

Added to Table 2 legend: Samples selected as “highly positive” for Spike and RBD levels were randomly selected samples at least 10 standard deviations greater than average negative control Spike OD value.

5. Line 216- What is the basis of 100% correlation? p-values would be useful here.

Added to Table 2 legend: For Positive Spike and Positive RBD samples, p values of Spike OD to RBD (<0.00001) and to Nucleocapsid protein (p<0.000001) were significant.

6. Line 308- reference?

Added reference 23 to bolster this.

7. Figure 1- Label X axis as OD units

Thank you, we changed the label on X axis.

8. Figure 2- Mention days/dates here. Or normalize the percent positivity by the no. of days for each month (especially for August and December)

We now mention dates in figure for all months, including August and December.

9. Supplementary Figure 1E- Label the monomeric peak

Added to Supplementary Figure 1: "The S protein monomer is predicted to sediment at approximately 5.5 S, but the exact value will depend on the hydrodynamic shape of the isolated monomeric species." …..and we did not detect it given the sensitivity range that shows the trimer in this assay.

10. Overall comment: what are the OD values in terms of antibody titers?

This is an interesting but difficult question to directly address, as only a surrogate S-protein monoclonal could be used, and that is arguably inadequate to model the true reactivity of a natural polyclonal antibody response based on content in human serum. More specifically, we could use a known human monoclonal against the S protein to generate a standard curve of antibody concentration versus OD values in the ELISA assay, so that we can then attempt to fit said curve to human serum readings. However, use of a known quantity of a human monoclonal antibody that recognizes a single epitope is not an accurate representation of OD values of mixed polyclonal reactivity against the entire S protein that likely binds dozens of antibodies at the same time on a stoichiometric basis. Thus, we did not employ this strategy as a surrogate for tittering OD values versus presumed antibody quantity.

11. Were serum IgM levels against SARS-CoV-2 measured in these individuals?

This is a good question, but we elected not to evaluate serum IgM levels in our donor samples, although this strategy is interesting as it would have potentially shown us individuals that were infected within the past 2-4 weeks (but not longer). The value of such temporal information through IgM analysis was deemed less critical given limited time and resources.

Reviewer #3: 

1. Line number 125/126 Board (IRB) reviewed the proposed SARS-CoV-2 serology initiative and classified it as non-human rearch. Please correct the sentence

Thank you, the type has been corrected.

2. The authors talked about significant variation in geographical analysis for past infection. Was comparison age matched as they observed age plays significant role so whether or not study population were matched for their age is important when considering geographical variations.

This is an interesting question although granular demographic information for these regions was not available so a comparison of regions of the GCMA by age groups or other demographic characteristics was not possible (unfortunately). We apologize for not being able to address this question with additional analyses, as we were unable to find a data source that should age prevalence within the population by region or zip code in the GCMA.

3. Authors should also discuss other studies with results showing variations in ethnic groups and should discuss their finding in comparison to CDC reports. They should discussion these studies too, if any done in the USA.

We thank the reviewer for this comment as it is an important issue and we have reformulated our discussion in lines 395-410 to discuss data from the CDC and past reports in the literature in comparison with our results related to race. Line 411-441 also deal more specifically with the total rate of past infectivity as detected by ELISA assays and why some of the differences in results may have occurred between select studies. There are also known rates of blood donation prevalence based on race, and we have cited those papers and also added another reference (#23), and in general, White non-Hispanics appear to be the predominant race that donates blood compared with other races when normalized to total population race percentages (this is now discussed and referenced, and reference #21 actually shows that blood donors are 90% White across more than 900,000 donors nationwide).

4. Detail statistical analysis for Table 4 will be helpful and a more explanatory table footnote will make it more clear to readers.

Reviewer 1 also mentioned this, we did more statistical analyses to bolster Table 4 (now Table 5) and added the p-values to table on the right-hand column. However, we have also added statistics and p values to table 2 in case the reviewer also meant that (added to the footnote).

5. For figure 1.B p value must be indicated on the histogram

This figure was incorrectly called out in the Results section as Fig 1B and should be Fig 2. P-value was added to Figure 2.

Responses to the Editors

 Done

2. In your Methods section, please provide additional information about the biological samples and the demographic details of the donors. Please ensure you have provided sufficient details to replicate the analyses such as:

a) the date(s) researchers accessed the samples,

To address this, we added information to lines 118-120

b) the date range that samples were originally collected,

and the date range the samples were collected was added to line 118-119.

c) a description of any inclusion/exclusion criteria that were applied to sample selection,

We excluded some samples for this study and added “Samples without complete biogeographical data were excluded” to lines 129-131 to address this, and samples that were duplicate donors.

d) a table of relevant demographic details of donors (if disclosed), \\\\.

We have included a Supporting Information table as an Excel spreadsheet of relevant demographic information sufficient to reproduce the data in this publication. While we have excluded zip code from the demographic data, we have replaced it with a more general region location to further de-identify the information that still allows full reproducibility of our analyses.

3. Please provide sequences or accession numbers for the SARS-CoV-2 antigens for S protein and RBD used in the study.

Vectors were received from Dr Florian Krammer. Added to Acknowledgement as resources that also gives accession number from those publications referenced:

We would like to thank Dr Florian Krammer at Mount Sinai Icahn School of Medicine for providing the mammalian expression vectors for the modified S protein and the RBD protein (9,10), which were produced under HHSN272201400008C and obtained through BEI Resources, NIAID, NIH: Spike Glycoprotein Receptor Binding Domain (RBD) from SARS-Related Coronavirus 2, Wuhan-Hu-1 with C-Terminal Histidine Tag, Recombinant from HEK293F Cells, NR-52366 and Spike Glycoprotein (Stabilized) from SARS-Related Coronavirus 2, Wuhan-Hu-1 with C-Terminal Histidine Tag, Recombinant from Baculovirus, NR-52308.

4. For studies involving humans categorized by race/ethnicity, age, disease/disabilities, religion, sex/gender, sexual orientation, or other socially constructed groupings, authors should:

1) Explicitly describe their methods of categorizing human populations,

2) Define categories in as much detail as the study protocol allows,

3) Update outmoded terms and potentially stigmatizing labels to more current, acceptable terminology.

Examples: “Caucasian” should be changed to “white”

For this study, donors who presented to Hoxworth Blood Center were in 6 racial categories; Asian, Black or African American, Native Hawaiian or Other Pacific Islander, White, Other, or Non-specified as delineated per NIH Notice Number: NOT-OD-15-089

5. Thank you for stating the following in the Financial Disclosure section:

"This study was supported the Howard Hughes Medical Institute (to J.D.M. No grant number, Molkentin Investigator). J.D.M was also supported by an internal grant from Cincinnati Children Research Foundation to conduct these studies on COVID-19 (Molkentin grant 1). Contributions to this work by P.S. were financially assisted by The John Hauck Foundation, Fifth Third Bank.

https://www.cincinnatichildrens.org/

https://fconline.foundationcenter.org/fdo-grantmaker-profile?key=HAUC002

https://www.hhmi.org/

 We note that you received funding from a commercial source: Fifth Third Bank.

The Fifth Third band reference for support was inaccurate and this has been removed in the revised manuscript. All other declarations are now made as requested in the sections "Competing interests" and "Data availability". We have also updated the "Sources of funding" section as requested with the website information, and proper declarations. The other requested information has been added to the cover letter to provide the requested assurances and to provide assurances as to the lack of "Competing interests". We have declared that no competing interests exist with this current work (all authors).

6. We noted in your submission details that a portion of your manuscript may have been presented or published elsewhere. [https://www.medrxiv.org/content/10.1101/2021.03.11.21253263v1.supplementary-material] Please clarify whether this publication was peer-reviewed and formally published. If this work was previously peer-reviewed and published, in the cover letter please provide the reason that this work does not constitute dual publication and should be included in the current manuscript.

As requested this is now mentioned in the cover letter and we state that this current work has not been peer-reviewed and formally published, although an earlier version of this work was uploaded to the medRxiv preprint server as a preliminary manuscript, as the editor indicates. This work does not constitute a dual publication because the preprint server version of the paper is not an official publication and it was not peer-reviewed. We make mention of this fact in the "Competing interest" section of the revised paper.

7. We note that you have indicated that data from this study are available upon request. PLOS only allows data to be available upon request if there are legal or ethical restrictions on sharing data publicly. For information on unacceptable data access restrictions, please see http://journals.plos.org/plosone/s/data-availability#loc-unacceptable-data-access-restrictions.

 We now include an Excel spreadsheet with as Supportive data that includes all the raw demographic data for the de-identified donor samples used in this study. This information is included in the cover letter as requested. Hence all raw and primary data are shared, and there is no full gel files of concern that is not already contained within the primary data (S1 Fig) 

 Minimum anonymized data set is provided. 

8. PLOS ONE now requires that authors provide the original uncropped and unadjusted images underlying all blot or gel results reported in a submission’s figures or Supporting Information files. This policy and the journal’s other requirements for blot/gel reporting and figure preparation are described in detail at https://journals.plos.org/plosone/s/figures#loc-blot-and-gel-reporting-requirements and https://journals.plos.org/plosone/s/figures#loc-preparing-figures-from-image-files. When you submit your revised manuscript, please ensure that your figures adhere fully to these guidelines and provide the original underlying images for all blot or gel data reported in your submission. See the following link for instructions on providing the original image data: https://journals.plos.org/plosone/s/figures#loc-original-images-for-blots-and-gels.

There is no full gel file of concern that is not already contained within the primary data (S1 Fig), hence Supporting Information is not needed. Indeed, the only gel image is from S1 Fig B and it is already the full raw data of the entire gel (not cropped), which is readily provided. All figures are in Tiff format as requested

9. We note that you have included the phrase “data not shown” in your manuscript. Unfortunately, this does not meet our data sharing requirements. PLOS does not permit references to inaccessible data. We require that authors provide all relevant data within the paper, Supporting Information files, or in an acceptable, public repository. Please add a citation to support this phrase or upload the data that corresponds with these findings to a stable repository (such as Figshare or Dryad) and provide and URLs, DOIs, or accession numbers that may be used to access these data. Or, if the data are not a core part of the research being presented in your study, we ask that you remove the phrase that refers to these data.

The reference to data not shown has been removed and we have modified this sentence and provided the necessary reference in the revised manuscript. Line 211-213 ELISA comparison between our ELISA and Krammer (9,10). 

10. Your ethics statement should only appear in the Methods section of your manuscript. If your ethics statement is written in any section besides the Methods, please move it to the Methods section and delete it from any other section. Please ensure that your ethics statement is included in your manuscript, as the ethics statement entered into the online submission form will not be published alongside your manuscript.

 Ethics statement has been moved to MM.

11. We note that Figure 2 in your submission contain map images which may be copyrighted. All PLOS content is published under the Creative Commons Attribution License (CC BY 4.0), which means that the manuscript, images, and Supporting Information files will be freely available online, and any third party is permitted to access, download, copy, distribute, and use these materials in any way, even commercially, with proper attribution. For these reasons, we cannot publish previously copyrighted maps or satellite images created using proprietary data, such as Google software (Google Maps, Street View, and Earth). For more information, see our copyright guidelines: http://journals.plos.org/plosone/s/licenses-and-copyright.

The original figure has been removed from the paper and an open source similar to the original has been added.

11.1. You may seek permission from the original copyright holder of Figure 2 to publish the content specifically under the CC BY 4.0 license. 

 11.2. If you are unable to obtain permission from the original copyright holder to publish these figures under the CC BY 4.0 license or if the copyright holder’s requirements are incompatible with the CC BY 4.0 license, please either i) remove the figure or ii) supply a replacement figure that complies with the CC BY 4.0 license. Please check copyright information on all replacement figures and update the figure caption with source information. If applicable, please specify in the figure caption text when a figure is similar but not identical to the original image and is therefore for illustrative purposes only.

The Map that is now provided was adapted from http://viewer.nationalmap.gov/viewer/ and is public domain.

Citation:

USGS the national Map: National Boundaries Dataset, 2DEP Elevation Program, Geographical Names Information System, National Hydrography Dataset, National land Cover Database, national Structures, dataset, and National Transportation Dataset; USGS Global Ecosystem; U.S. Census Bureau TIGER/Line data; USFS Road Data; Natural Earth data; U.S. Department of State Humanitarian Information Unit; and NOAA National Centers for Environmental Information, US Coastal Relief Model. Data refreshed May, 2020.

12. Please include captions for your Supporting Information files at the end of your manuscript, and update any in-text citations to match accordingly. Please see our Supporting Information guidelines for more information: http://journals.plos.org/plosone/s/supporting-information.

This has been added.

---

## [Decision Letter · Decision Letter 1]

1 Jul 2021

Seroprevalence of SARS-CoV-2 Infection in Cincinnati Ohio USA from August to December 2020

PONE-D-21-09100R1

Dear Dr. Molkentin,

We’re pleased to inform you that your manuscript has been judged scientifically suitable for publication and will be formally accepted for publication once it meets all outstanding technical requirements.

Kind regards,

Binod Kumar, PhD

Academic Editor

PLOS ONE

Additional Editor Comments (optional):

Reviewers' comments:

Reviewer's Responses to Questions

**Comments to the Author**

1. If the authors have adequately addressed your comments raised in a previous round of review and you feel that this manuscript is now acceptable for publication, you may indicate that here to bypass the “Comments to the Author” section, enter your conflict of interest statement in the “Confidential to Editor” section, and submit your "Accept" recommendation.

Reviewer #1: All comments have been addressed

Reviewer #2: All comments have been addressed

Reviewer #3: All comments have been addressed

2. Is the manuscript technically sound, and do the data support the conclusions?

Reviewer #1: Yes

Reviewer #2: Yes

Reviewer #3: Yes

3. Has the statistical analysis been performed appropriately and rigorously? 

Reviewer #1: N/A

Reviewer #2: Yes

Reviewer #3: Yes

4. Have the authors made all data underlying the findings in their manuscript fully available?

Reviewer #1: Yes

Reviewer #2: Yes

Reviewer #3: Yes

5. Is the manuscript presented in an intelligible fashion and written in standard English?

Reviewer #1: Yes

Reviewer #2: Yes

Reviewer #3: Yes

6. Review Comments to the Author

Reviewer #1: (No Response)

Reviewer #2: Thanks for addressing the comments. The manuscript is acceptable for publication in the present format.

Reviewer #3: Authors have addressed all queries and answered them satisfactorily. The research article can be accepted for the publication

7. PLOS authors have the option to publish the peer review history of their article (what does this mean?). If published, this will include your full peer review and any attached files.

Reviewer #1: **Yes: **Rupkatha Mukhopadhyay

Reviewer #2: No

Reviewer #3: No

---

## [Editor Report · Acceptance letter]

2 Jul 2021

PONE-D-21-09100R1 

Seroprevalence of SARS-CoV-2 infection in Cincinnati Ohio USA from August to December 2020 

Dear Dr. Molkentin:

I'm pleased to inform you that your manuscript has been deemed suitable for publication in PLOS ONE. Congratulations! Your manuscript is now with our production department. 

Kind regards, 

on behalf of

Dr. Binod Kumar 

Academic Editor

PLOS ONE